# Functional recurrent laryngeal nerve regeneration using a silicon tube containing a collagen gel in a rat model

**Ryohei Asai**[1]*, **Sohei Ishii**[1], **Ikuo Mikoshiba**[1], **Tomohiko Kazama**[2], **Hiroumi Matsuzaki**[1], **Takeshi Oshima**[1], **Taro Matsumoto**[2]

**1** Department of Otolaryngology-Head and Neck Surgery, Nihon University School of Medicine, Tokyo, Japan, **2** Department of Functional Morphology, Division of Cell Regeneration and Transplantation, Nihon University School of Medicine, Tokyo, Japan

* asai.ryohei@nihon-u.ac.jp

**Data Availability Statement:** All relevant data are within the paper and its Supporting Information files.

## Abstract

In this study, we examined the effect of differing gap lengths on regeneration of transected recurrent laryngeal nerves using silicon tubes containing type I collagen gel and the ability of this regeneration to result in restoration of vocal fold movements in rats. We simulated nerve gaps in Sprague-Dawley rats by transecting the left recurrent laryngeal nerves and bridged the nerve stumps using silicon tubes containing type 1 collagen gel. Three experimental groups, in which the gap lengths between the stumps were 1, 3, or 5 mm, were compared with a control group in which the nerve was transected but was not bridged. After surgery, we observed vocal fold movements over time with a laryngoscope. At week 15, we assessed the extent of nerve regeneration in the tube, histologically and electrophysiologically. We also assessed the degree of atrophy of the thyroarytenoid muscle (T/U ratio). Restoration of vocal fold movements was observed in 9 rats in the 1-mm group, in 6 rats in the 3-mm group, and in 3 rats in the 5-mm group. However, in most rats, restoration was temporary, with only one rat demonstrating continued vocal fold movements at week 15. In electromyograph, evoked potentials were observed in rats in the 1-mm and 3-mm groups. Regenerated tissue in the tube was thickest in the 1-mm group, followed by the 3-mm and 5-mm groups. The regenerated tissue showed the presence of myelinated and unmyelinated nerve fibers. In assessment of thyroarytenoid muscle atrophy, the T/U ratio was highest in the 1-mm group, followed by the 3-mm and 5-mm groups. We successfully regenerated the nerves and produced a rat model of recurrent laryngeal nerve regeneration that demonstrated temporary recovery of vocal fold movements. This rat model could be useful for assessing novel treatments developing in the future.

## Introduction

Peripheral nerves typically have high regenerative capacity, unlike central nerves. Therefore, the axons of injured peripheral nerves regenerate following Wallerian degeneration, with

**Funding:** This research was supported by the JSPS KAKENHI Grant Numbers 17H04152 [MT] and 17K11411 [IS], by AMED-supported Program for the Research Project for Practical Applications of Regenerative Medicine (18bk0104005h001)[MT], and by Nihon University President Grant Initiative (2018-2020)[MT]. The funders had no role in study design, data collection and analysis, decision to publish, or preparation of the manuscript.

**Competing interests:** The authors have declared that no competing interests exist.

Schwann cells playing a central role in the process. However, when the peripheral nerve is severed at the site of injury, nerve regeneration is rendered impossible and may require surgery. Direct suturing of the nerve stumps is a standard treatment modality, whenever possible. However, when the nerve gap is too large to be closed by direct suturing, the gap must be bridged to re-establish the continuity of the nerve. In such cases, autologous nerve grafting is the conventional treatment modality [1], although neuropathy at the nerve harvest site poses a problem. In addition, the sites and lengths of nerves that can be harvested are limited. A recently developed alternative to nerve grafting is nerve guidance conduit, which can be grafted at the site of nerve gap to serve as a scaffold for nerve regeneration. Nerve guidance conduits have been principally used in sensory nerves to achieve regeneration without sacrificing a functional nerve [2].

Thyroid, pharyngeal, and/or laryngeal cancers, which have become more prevalent in recent years, are primarily treated with surgical resection. Although surgical resection can result in complete cure, postoperative complications are often observed. One such complication is postoperative nerve palsy, which affects the quality of life and can hamper continued medical care. A typical example of postoperative nerve palsy in the head and neck region is the recurrent laryngeal nerve palsy. The recurrent laryngeal nerve is the main nerve responsible for vocal fold movements. Therefore, recurrent laryngeal nerve palsy impairs vocal fold movements, resulting in symptoms such as hoarseness, dysphagia, and dyspnea [3]. Unavoidable transection of the recurrent laryngeal nerve, often caused by surgical maneuvers, is repaired with treatments such as nerve suturing. Although nerve suturing can prevent the atrophy of laryngeal muscles, vocal fold movements are not restored [4]. Hence, novel modes of treatment that target restoration of function are sought.

In many cases, vocal fold movements are restored after injury to the recurrent laryngeal nerve if the injury is mild. An animal experiment showed that vocal fold movements can be restored, if the injury did not involve tearing [5]. However, if a nerve is torn, vocal fold movements almost never recover, even with direct suturing. Several theories have been posited to explain the absence of restoration of vocal fold movements even with anastomotic reconstruction [6]. One of the most persuasive theories is misdirection, a phenomenon in which the transected nerve fibers project toward nerve fibers other than the original, during the stage of nerve regeneration [7]. During the regeneration of a mixed motor, sensory, and autonomic nerve such as the recurrent laryngeal nerve, misdirection occurs when the transected motor nerve projects toward the distal motor, distal sensory, or distal autonomic nerve stumps, all of which have different actions.

According to recent studies, if a tear in a peripheral nerve results in a short gap, a nerve guidance conduit yields more favorable results than nerve anastomosis [8]. As has been demonstrated, direct anastomosis of the recurrent laryngeal nerve stumps fails to restore vocal fold movements. Therefore, research using various types of nerve guidance conduits are likely to be performed in the near future.

Several previous studies have attempted to regenerate the transected recurrent laryngeal nerves using nerve guidance conduits primarily in animals such as rats. The results of several experiments performed using different tube materials, fillings, growth factors, and cells, have introduced varied types of conduits [9–15]. However, these studies were performed in different gap lengths between the recurrent laryngeal nerve stumps, and no study has examined the differences in outcomes with differing gap lengths. In addition, few studies have reported restoration of vocal fold movements. Therefore, we analyzed the effect of differing gap lengths on regeneration of transected recurrent laryngeal nerves using silicon tubes of different lengths containing type I collagen gel and the ability of this regeneration to result in restoration of vocal fold movements in rats.

## Materials and methods

### Laboratory animals and ethical considerations

The experiment was conducted with 8-week-old male Sprague-Dawley rats (Oriental Yeast Co. Ltd., Tokyo, Japan). This study was carried out in strict accordance with the guidelines of the Nihon University School of Medicine for animal experiments. This study was approved by the Nihon University Animal Care and Use Committee (AP16M018, AP16M043). All surgeries were performed under general anesthesia, and all efforts were made to minimize suffering. Animals were maintained on standard rodent chow and sufficient water. After surgery, aspirin (100 mg/kg/day) was administered for 7 days for analgesia and Cefalexin (120 mg/kg/day) was administered for 5 days for prevention of infection. $CO_2$ gas was used for sacrifice. The wound was monitored postoperatively, and weekly weight measurements were performed. We decided to perform sacrifice with $CO_2$ gas as a humane endpoint in case of infection or weight loss of more than 20% in one week.

### Preparation of recurrent laryngeal nerve regeneration models

The experiments were performed in rats under general anesthesia induced via intraperitoneal injections of 0.15 mg/kg body weight medetomidine chloride, 2 mg/kg body weight midazolam, and 2.5 mg/kg body weight butorphanol tartrate. The surgery was performed under a stereomicroscope (SZ2-ILST, Olympus, Tokyo, Japan). A vertical skin incision was made in the anterior part of the neck, and the muscles in the anterior part of the neck were reflected in a lateral direction to expose the trachea. The connective tissue surrounding the trachea was detached until the left and right recurrent laryngeal nerves were visualized, and the left recurrent laryngeal nerve was transected at the level of the seventh tracheal ring. Silicon tubes 0.5 mm in internal diameter (Laboran tube, AS ONE Corp., Osaka, Japan) were filled with type I collagen gel (Cellmatrix, Nitta Gelatin Inc., Osaka, Japan). The nerve stumps were drawn into the tube from both ends, 0.5 mm at a time, and were secured using 10–0 nylon sutures. The silicon tubes containing type I collagen gel were heated to a temperature of 37˚C prior to use.

### Experimental protocol

The experiment was performed with 37 rats, which were divided into three groups based on the gap length between the transected nerve stumps: a 1-mm group (n = 11), a 3-mm group (n = 11), and a 5-mm group (n = 11). In each group, 1 mm, 3 mm, or 5 mm of recurrent laryngeal nerve was transected to create 1-mm, 3-mm, or 5-mm gaps between stumps, respectively, which were stabilized using silicon tubes. We also formed a control group (n = 4) in which the left recurrent laryngeal nerve was transected but a silicon tube was not used. In the control group, 5-mm segments of the recurrent laryngeal nerves were transected, and the stumps were ligated using silk threads (Fig 1).

All rats (n = 37) were assessed via laryngoscopy immediately following treatment (0 w) and at 5 weeks (5 w), 10 weeks (10 w), and 15 weeks (15 w) after treatment. Evoked electromyography was performed at 15 w in rats in the control group (n = 2) and in those in the bridged groups (n = 3 each). Histological examinations of the tissues in the tubes were conducted at 15 weeks in rats in the bridged groups (n = 11 each), while histological examinations of the intrinsic laryngeal muscles were conducted in rats in the control group (n = 2) and in those in the bridged groups (n = 8 each). The insertion of the electrode of evoked electromyography injures the intrinsic laryngeal muscles. Therefore, different rats were used for evoked electromyography and histological examination of intrinsic laryngeal muscles.

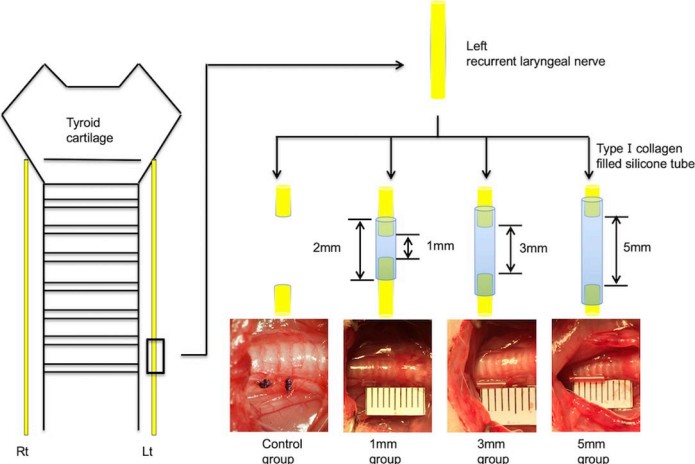

**Fig 1. Rat model of recurrent laryngeal nerve regeneration.** Sprague-Dawley rats (n = 37) divided into a control group (n = 4) and a bridged group (n = 33). This bridged group was further divided into a 1-mm gap group, a 3-mm gap group, and a 5-mm gap group (n = 11 each). In the bridged groups, nerve stumps were drawn 0.5 mm at a time from both ends into silicon tubes of different lengths and filled with type I collagen gel. In the control group, the nerve stumps were ligated using a thread.

## Laryngoscopy

The vocal fold movements in rats were assessed using a rigid endoscope (TrueView II; Olympus, Tokyo, Japan). After the induction of general anesthesia in rats via intraperitoneal injections of 0.15 mg/kg body weight medetomidine chloride, 2 mg/kg body weight midazolam, and 2.5 mg/kg body weight butorphanol tartrate, the rats were secured to an intubation stand (KN-1014; Natsume Seisakusho Co., Ltd., Tokyo, Japan). The vocal fold movements were assessed using the rigid endoscope by pulling the tongue forward. The assessments were performed immediately and at 5 w, 10 w, and 15 w after treatment. The rat larynx is characterized by small vocal cords and large arytenoid cartilages. We, therefore, assessed vocal fold movements by observing the mobility of the arytenoid cartilage. A normal arytenoid cartilage adducts during expiration and abducts during inspiration. Therefore, we compared the movements of the arytenoid cartilage on the treated side to those on the untreated side and scored the differences in movement patterns on a five-point scale, based on the criteria described be Tessema et al [5]. The criteria are as follows: 0 = no vocal fold movement; 1 = slight vocal fold movements; 2 = < 50% abduction of vocal folds; 3 = > 50% abduction, unsmooth vocal fold movements; 4 = vocal fold movements and positions indistinguishable from normal (Fig 2). After the blinding of the formed groups, the movements were scored by three otolaryngologists who were not directly involved in the study.

## Evoked electromyography

The recurrent laryngeal nerve was stimulated, and the evoked potentials in the thyroarytenoid muscle were measured. At 15 w, after general anesthesia induction in rats via intraperitoneal injections, a vertical skin incision was made in the anterior part of the neck to expose the left and right recurrent laryngeal nerves. After removing the silicon tube, bipolar hook-type electrodes (EKM2-5050; Nihon Bioresearch Inc., Aichi, Japan) were attached as stimulating electrodes to the left recurrent laryngeal nerve (i.e. the treated side) at proxymal of the bridged site, while needle electrodes (EKB-13005K; Nihon Bioresearch Inc., Aichi, Japan) were inserted as recording electrodes into the thyroarytenoid muscle on the treated side. On the untreated

| Expiration | Inspiration | Pattern | Score |
|---|---|---|---|
| | | No movement | 0 |
| | | Slight movement | 1 |
| | | <50% abduction | 2 |
| | | >50% abduction | 3 |
| | | Indistinguishable from normal | 4 |

**Fig 2. Scores for vocal fold movements.** Vocal fold movements are assessed on a five-point scale based on the criteria described by Tessema et al. The position of the arytenoid cartilage during expiration and inspiration is marked with a white-dotted line and a yellow-dotted line, respectively. 0 = no vocal fold movement; 1 = slight vocal fold movements; 2 = < 50% abduction of vocal folds; 3 = > 50% abduction, unsmooth vocal fold movements; 4 = vocal fold movements and positions indistinguishable from normal.

side, electrodes were placed at the same levels as those on the treated side. The indifferent electrode was inserted into the thigh muscles. An electrical stimulator (FE180; ADInstruments, Dunedin, New Zealand) was used to induce supramaximal stimulation (1.0–1.5 A) at intervals of 0.1 ms. Compound muscle-action potentials were recorded using PowerLab and LabChart6 (ADInstruments).

## Histological examination

We removed the silicon tube and determined macroscopically whether the nerve stumps were connected by regenerated nerve tissue. The tissues were removed from the tube and were subjected to primary stabilization for 4 hours in 2.5% glutaraldehyde in 0.1 M cacodylate buffer solution (pH 7.4). After washing the tissues using 0.1 M cacodylate buffer solution (pH 7.4), secondary stabilization using 1% osmium tetroxide in 0.1 M cacodylate buffer solution (pH 7.4) was performed. The tissues were dehydrated using increasing concentrations of ethanol and were embedded in Epon resin (Epon812; TAAB Laboratories Equipment Ltd., Berks, UK). Semi-thin and thin sections were obtained using an Ultramicrotome (Ultracut UCT, Leica, Wetzlar, Germany), were stained using toluidine blue, and were observed under an optical microscope (BZ-x710, Keyence Corporation, Osaka, Japan). The sections were double electron-stained using uranyl acetate and lead stain solution and were observed under a transmission-electron microscope (JEM-1200EX, JEOL Ltd., Tokyo, Japan). Sections were prepared from the centers of the regenerated tissues.

Histological examinations of intrinsic laryngeal muscles were performed. After the rats were euthanized, their larynges were removed, fixed in 10% neutral buffered formalin, and embedded in paraffin. Frontal sections were prepared at levels where arytenoid cartilages could be observed, stained using hematoxylin-eosin, and imaged using an optical microscope. We selected the sections in which the left and right arytenoid cartilages were the most symmetrical and measured the cross-sectional areas of the left and right thyroarytenoid muscles. The

total area of thyroarytenoid muscle fibers was measured after they were binarized using the image-analysis software (ImageJ ver. 1.51; National Institutes of Health, Rockville, MD). The ratio of the area on the treated side to the area on the untreated side (T/U ratio) was calculated as a form of quantitative assessment.

## Statistical assessment

The restoration of vocal fold movements in all groups at all points in time was analyzed statistically using the chi-square test. Results with P < 0.05 were considered statistically significant.

Data for diameters of tissues in tubes and T/U ratios for thyroarytenoid muscles were statistically analyzed using one-way analysis of variance (ANOVA), and the Tukey's multiple comparisons test was used as a post-hoc test. Results with P < 0.05 were considered statistically significant.

## Results

### Assessment of vocal fold movements

We assessed the restoration of vocal fold movements using an endoscope immediately and 5 weeks, 10 weeks, and 15 weeks after treatment. The results of this assessment are shown in Table 1.

The rats who demonstrated vocal fold movements at each point in time were represented as ratio (%) of movements. Vocal fold movements observed immediately after surgery on the treated side disappeared in all examined rats. In the control group, none of the rats demonstrated restoration of vocal fold movements during the 15-week observation period. However, vocal fold movements were restored in some rats in all three bridged groups. At 5 w, vocal fold movements were observed in 9 rats in the 1-mm group, in 4 rats in the 3-mm group, and in 3 rats in the 5-mm group. At 10 w, these numbers declined to 3 rats in the 1-mm group, 3 rats in the 3-mm group, and 1 rat in the 5-mm group. At 15 w, these numbers further declined to 1 rat each in all the 3 groups. At 5 w, the number of rats that demonstrated restoration of vocal

**Table 1. Scores for vocal fold movements in each group.**

| control | | | | | 1mm group | | | | | 3mm group | | | | | 5mm group | | | | |
|---|---|---|---|---|---|---|---|---|---|---|---|---|---|---|---|---|---|---|---|
| No. | 0w | 5w | 10w | 15w | No. | 0w | 5w | 10w | 15w | No. | 0w | 5w | 10w | 15w | No. | 0w | 5w | 10w | 15w |
| #1 | 0 | 0 | 0 | 0 | #1 | 0 | 0 | 4 | 4 | #1 | 0 | 0 | 3 | 3 | #1 | 0 | 2 | 3 | 3 |
| #2 | 0 | 0 | 0 | 0 | #2 | 0 | 3 | 3 | 0 | #2 | 0 | 3 | 0 | 0 | #2 | 0 | 2 | 0 | 0 |
| #3 | 0 | 0 | 0 | 0 | #3 | 0 | 2 | 3 | 0 | #3 | 0 | 3 | 0 | 0 | #3 | 0 | 2 | 0 | 0 |
| #4 | 0 | 0 | 0 | 0 | #4 | 0 | 3 | 0 | 0 | #4 | 0 | 1 | 1 | 0 | #4 | 0 | 0 | 0 | 0 |
| Ratio(%) of movement | 0/4 | 0/4 | 0/4 | 0/4 | #5 | 0 | 2 | 0 | 0 | #5 | 0 | 1 | 0 | 0 | #5 | 0 | 0 | 0 | 0 |
| | (0) | (0) | (0) | (0) | #6 | 0 | 2 | 0 | 0 | #6 | 0 | 0 | 2 | 0 | #6 | 0 | 0 | 0 | 0 |
| | | | | | #7 | 0 | 1 | 0 | 0 | #7 | 0 | 0 | 0 | 0 | #7 | 0 | 0 | 0 | 0 |
| | | | | | #8 | 0 | 1 | 0 | 0 | #8 | 0 | 0 | 0 | 0 | #8 | 0 | 0 | 0 | 0 |
| | | | | | #9 | 0 | 1 | 0 | 0 | #9 | 0 | 0 | 0 | 0 | #9 | 0 | 0 | 0 | 0 |
| | | | | | #10 | 0 | 0 | 0 | 0 | #10 | 0 | 0 | 0 | 0 | #10 | 0 | 0 | 0 | 0 |
| | | | | | #11 | 0 | 0 | 0 | 0 | #11 | 0 | 0 | 0 | 0 | #11 | 0 | 0 | 0 | 0 |
| | | | | | Ratio(%) of movement | 0/11 | 9/11* | 3/11 | 1/11 | Ratio(%) of movement | 0/11 | 4/11 | 3/11 | 1/11 | Ratio(%) of movement | 0/11 | 3/11 | 1/11 | 1/11 |
| | | | | | | (0) | (81.8) | (27.3) | (9.1) | | (0) | (36.4) | (27.3) | (9.1) | | (0) | (27.3) | (9.1) | (9.1) |

*: p<0.05 V.S. 5w in 5mm group (chi-squared test)

fold movements was significantly higher in the 1-mm group than that in the control group, 3-mm group, and 5 -mm group (P < 0.05). However, in many cases, restoration of vocal fold movement was temporary; some rats that demonstrated movements at 5 w or 10 w did not demonstrate movements at 15 w. At the final observation (15 w), vocal fold movements were evident in only one rat in each bridged group. Only 1 rat in the 1-mm group was scored as 4, which represented improvement in movements to the same extent as that on the untreated side.

## Assessment of regenerated tissue in tubes

Histological examination of the regenerated tissue in the tubes was performed to assess nerve regeneration. After 15 w, we removed the silicon tubes bridging the nerve stumps of the transected recurrent laryngeal nerve under general anesthesia and assessed the regenerated tissue inside the tubes. In all rats, regenerated tissue connected the nerve stumps of the transected recurrent laryngeal nerve, and the findings confirmed that the continuity of the nerve was restored. Macroscopically, center of the regenerated tissue was constricted; while the tissue was thickest in the 1-mm group, followed by the 3-mm and 5-mm groups (Fig 3). Observation of the center of the regenerated tissues under an optical microscope revealed the presence of fibroblasts surrounding the tissues and regenerated nerve fibers and neovessels near the center of the tissue. The mean diameters of the centers of the regenerated tissues in the tubes in the 1-mm, 3-mm, and 5-mm groups were 361.9 ± 55.6 μm (mean ± standard deviation), 258.2 ± 62.4 μm, and 206.2 ± 58.5 μm, respectively. Thus, the tissues were significantly thinner in the 3-mm and 5-mm groups than those in the 1-mm group (p < 0.01 and p < 0.005, respectively) (Fig 4). Observation of the tissue under an electron microscope revealed the presence of myelinated and unmyelinated nerves in all groups (Fig 5).

## Assessment of laryngeal muscles

We histologically assessed the intrinsic laryngeal muscles that are innervated by the recurrent laryngeal nerves to assess the functional vocal fold movements. After the rats were euthanized, their larynges were removed, fixed in formalin, and embedded in paraffin to obtain frontal sections. Of all the intrinsic laryngeal muscles that are innervated by the recurrent laryngeal nerve, we observed the thyroarytenoid muscle, as it accurately represented the intrinsic laryngeal muscles. In all the examined rats, the thyroarytenoid muscle on the treated side showed signs of atrophy, while that on the untreated site did not. Observation under high magnification also revealed the atrophy of muscle fibers (Fig 6). The T/U ratios in the 1-mm, 3-mm, and 5-mm groups were 76.4 ± 5.1% (mean ± standard deviation), 69.6 ± 6.1%, and 63.6 ± 6.0%,

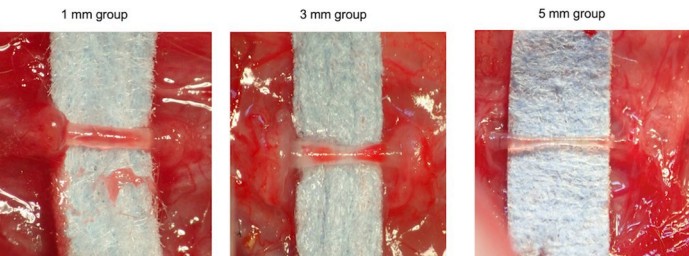

**Fig 3. Macroscopic appearance of regenerated tissue in the tubes.** At 15 weeks, we euthanized the rats, removed the silicon tubes, and macroscopically examined the regenerated tissues. The regenerated tissue connected the nerve stumps in all the examined rats. The center of the regenerated tissue was constricted. The tissue was thickest in the 1-mm group, followed by the 3-mm and 5-mm groups.

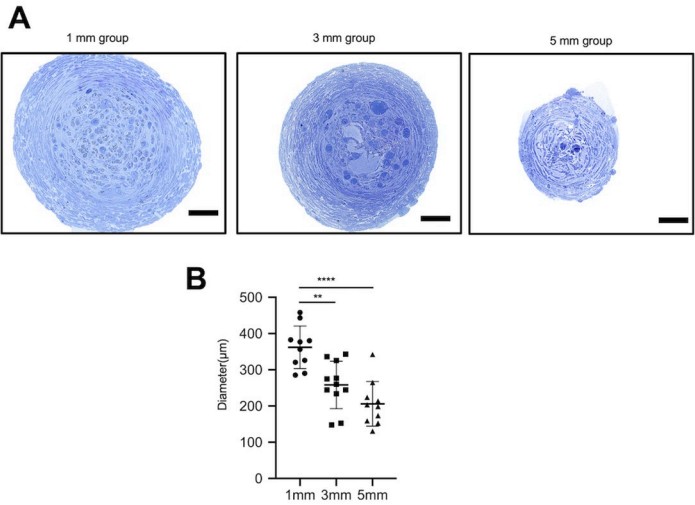

**Fig 4. Appearance of regenerated tissue in tubes under an optical microscope. (A)** At 15 weeks, the regenerated tissue inside the tubes was removed, and samples obtained from the centers of the tissues were prepared (toluidine blue staining). The tissue is surrounded by fibroblasts, and regenerated nerve fibers are observed in the center of the tissue. Scale bar: 50 μm. **(B)** We measured and compared the diameters of the regenerated tissues in the tubes. The diameter of the regenerated tissue is significantly more in rats in the 1-mm group than that in those in the 3-mm or 5-mm groups. ** p < 0.01, **** p < 0.001 (one-way ANOVA, Tukey's multiple comparisons test).

respectively. Thus, muscle atrophy was the least severe in the 1-mm group, followed by the 3-mm and 5-mm groups, with a significant difference between the 1-mm and 5-mm groups (P < 0.01) (Fig 7).

## Assessment of evoked electromyograms

We performed evoked electromyography to assess whether the regenerated part of the recurrent laryngeal nerve had electrophysiologically connected with the transected stumps. After 15 w, we electrically stimulated the recurrent laryngeal nerves on the treated and untreated sides and analyzed the evoked electromyograms of the thyroarytenoid muscle on both sides. In all rats in the control group, no evoked potentials were observed on the treated side. Among rats in the bridged groups, evoked potentials were detected in 2 of 3 rats in the 1-mm group and in 1 of 3 rats in the 3-mm group. Further, the compound muscle-action potential was smaller on

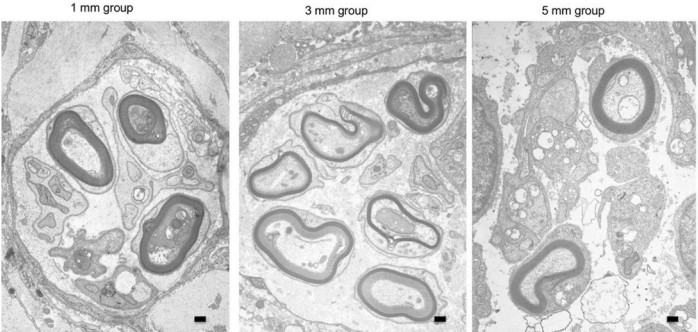

**Fig 5. Appearance of regenerated tissue in tubes under an electron microscope.** We prepared sections of regenerated tissues in tubes 15 weeks after treatment and observed the sections under an electron microscope. We observed the presence of myelinated and unmyelinated nerves. Scale bar: 500 nm.

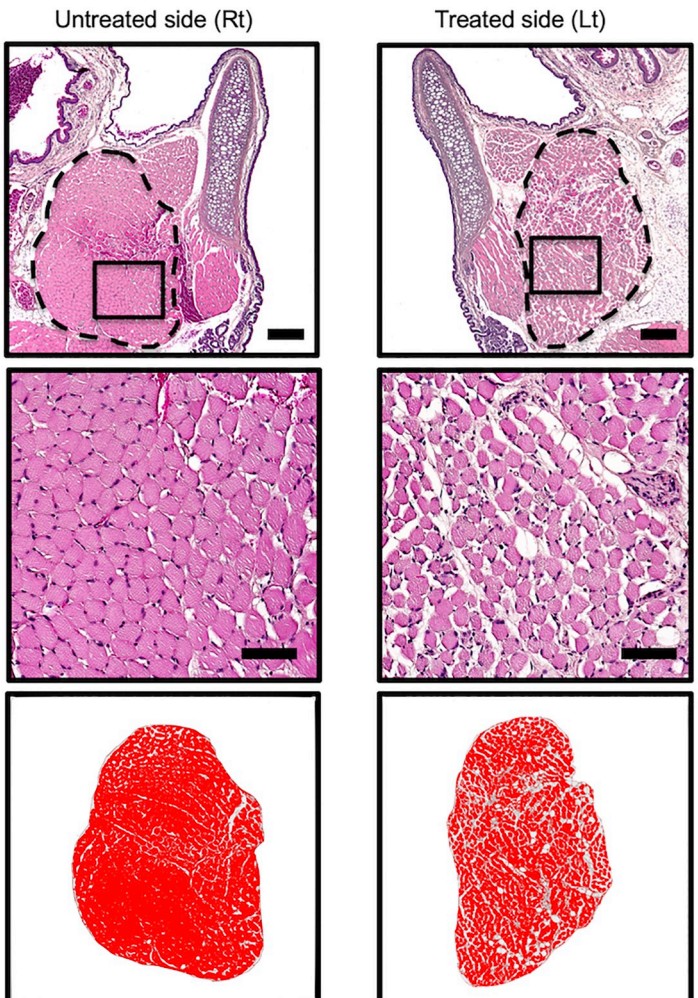

**Fig 6. Microscopic evaluation of thyroarytenoid muscle.** (Upper row) Low magnification: The thyroarytenoid muscle, in which the muscle fiber cross-sectional area is measured, is surrounded by a dotted line. Thyroarytenoid muscle atrophy is more severe on the treated side than that on the untreated side. Thyroarytenoid muscle on the treated side atorophied compared on the untreated side. Scale bar: 300 μm. (Middle row): High magnification: Magnified images of the boxes in the upper row. Atrophy of single muscle fibers can be observed. Scale bar: 100 μm (Lower row): The binarized thyroarytenoid muscle, and T/U ratios.

the treated side than on the untreated side in evoked myograms in the 1-mm and 3-mm groups, and the period of latency was longer. No statistically significant difference was evident in potentials between the 1-mm and 3-mm groups; however, the period of latency was longer in rats in the 3-mm group than that in those in the 1-mm group. In contrast, evoked potentials were not detected in any of the 3 rats analyzed in the 5-mm group (Fig 8).

## Examination of rats that demonstrated temporary restoration of vocal fold movements

We analyzed the data to determine the cause of temporary restoration of vocal fold movements that subsequently disappeared in some rats. Fig 9 shows a representative example of a rat in the 1-mm group (#3) that demonstrated temporary restoration of vocal fold movements. In this rat, vocal fold movements on the treated side had disappeared at 0 w after surgery, had

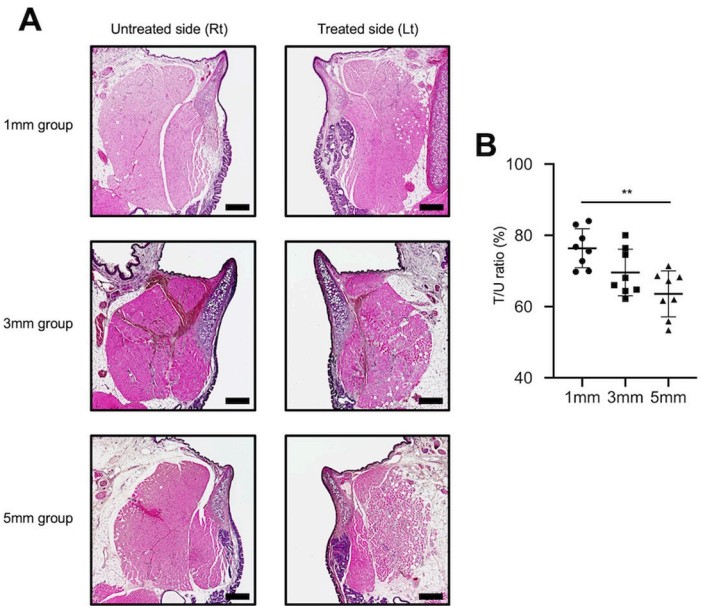

**Fig 7. Typical images of thyroarytenoid muscle tissue on the treated side in all groups. (A)** The degree of atrophy of the thyroarytenoid muscle in all groups was assessed using hematoxylin and eosin staining. All examined rats demonstrated overall atrophy of the thyroarytenoid muscle on the treated side. (B) The T/U ratios in the 1-mm, 3-mm, and 5-mm groups were 76.4 ± 5.1%, 69.6 ± 6.1%, and 63.6 ± 6.0%, respectively. Thus, atrophy was the least severe in the 1-mm group, followed by the 3-mm and 5-mm groups, with a significant difference between the 1-mm and 5-mm groups. ** p < 0.01, (one-way analysis of variance, Tukey's multiple comparisons test).

been restored by 5 w, were maintained by 10 w, and had disappeared by 15 w (Fig 9A). Histological examination of the regenerated tissue in the silicon tube revealed the presence of regenerated myelinated and unmyelinated nerves (Fig 9B and 9C). In evoked electromyography, presence of treatment-side recurrent laryngeal nerve-mediated action potentials was confirmed (Fig 9D). The above findings suggest that restoration of vocal fold movements could be

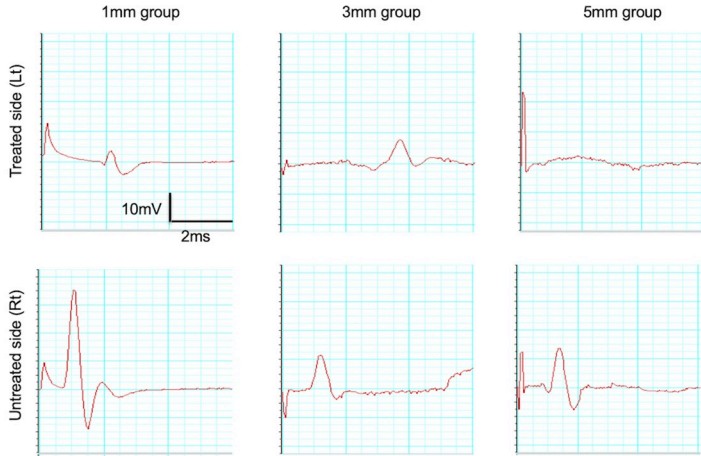

**Fig 8. Evoked electromyograms.** Typical electromyograms of rats in the 1-mm, 3-mm, and 5-mm groups. (Upper row): Treated side; (Lower row): Untreated side. In the 5-mm group, no potentials were evoked on the treated side. In the 1-mm and 3-mm groups, although potentials were evoked on the treated side, they were smaller than the action potentials on the untreated side, and the period of latency was longer than that on the untreated side.

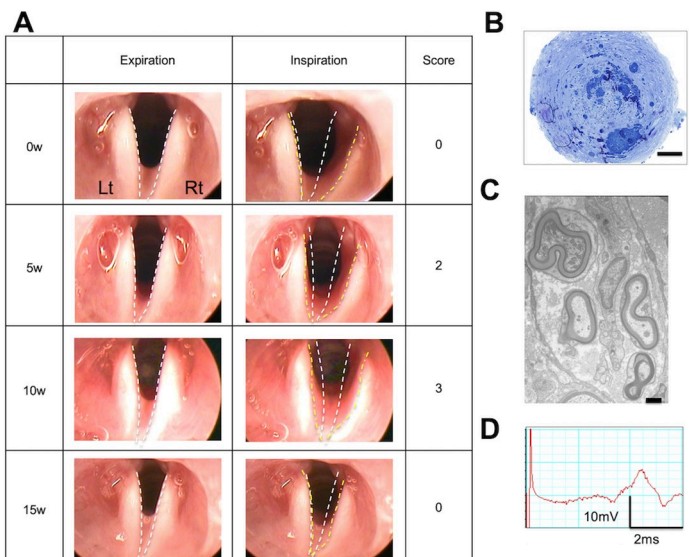

**Fig 9. Typical case demonstrating the temporary restoration of vocal fold movements (1-mm group, #3).** (A) Time series of vocal fold movements (B) Optical microscope images of regenerated tissue (scale bar: 50 μm) (C) Electron microscope images (scale bar: 500 nm) (D) Evoked electromyogram. (A) Restoration of vocal fold movements observed at 5 weeks and 10 weeks, but disappeared at 15 weeks. (B) Optical microscope images and (C) Electron microscope images show nerve regeneration. (D) Observed evoked potentials.

temporary, even if recurrent laryngeal nerve regeneration is confirmed electrophysiologically and histologically.

## Discussion

In this study, we used silicon tubes containing type I collagen gel to bridge gaps between the transected recurrent laryngeal nerve stumps, thereby regenerating the nerve. We conducted both functional and histological examinations.

Endoscopic assessment revealed that vocal fold movements recovered in some rats in all bridged groups.

In a previous study, which analyzed the regeneration of severed recurrent laryngeal nerves with gaps between the stumps in rats, the use of a self-assembling peptide led to restoration of vocal fold movements in 1 of 5 rats [15]. However, in most studies, vocal fold movements were not restored despite nerve fiber connections [10–12]. A recent molecular study elucidated that collagen promotes axonal regeneration in peripheral nerves [16]. In our study, unlike in other studies, type I collagen gel, kept heated at 37˚C, was used that acted as a scaffold and conceivably facilitated the extension of axons. Macroscopically, it was determined that regenerated tissue connected the severed nerve stumps in all rats. The regenerated tissue was thickest in the 1-mm group, followed by the 3-mm and 5-mm groups. Histological examination revealed the presence of myelinated and unmyelinated nerves in all regenerated tissues. Silicon tubes are strong and prevent the infiltration of granulation tissue. However, as silicon is non-absorbable, blood vessels do not penetrate the tube from outside, implying that the supply of nutrients and oxygen may depend on extension of blood vessels from the nerve stumps inside the tube. Therefore, regenerated tissue may be narrower in diameter when gaps are longer.

Assessment of laryngeal tissue revealed atrophy of the thyroarytenoid muscle in all rats at 15 w in terms of cross-sectional area of the entire muscle. This atrophy was the least severe in the 1-mm group, followed by the 3-mm and 5-mm groups, with a significant difference

between the 1-mm and 5-mm groups. In a study performed in rats with transected recurrent laryngeal nerves, atrophy of the thyroarytenoid muscle was speculated to begin 2 weeks after transection [17] and continued to atrophy significantly until 10 weeks, but no changes were evident after that point [18]. In a study involving nerve-muscle pedicle implantation with ansa cervicalis following transection of the recurrent laryngeal nerve, re-innervation led to recovery of atrophied laryngeal muscles at 10 weeks [19]. In this study, in the 1-mm group, the regenerated nerve rapidly established contact with the target laryngeal muscle, leading to early re-innervation, presumably alleviating laryngeal muscle atrophy and promoting recovery. This could be the reason for significantly less atrophy in rats the 1-mm group than that in the 5-mm group.

In this study, examination of rats demonstrating restored vocal fold movements revealed that despite histological and electrophysiological regeneration of the recurrent laryngeal nerve, restoration of vocal fold movements was temporary in most rats. One conceivable explanation could be early re-innervation of the posterior cricoarytenoid muscle and subsequent antagonist motor innervation. The posterior cricoarytenoid muscle, which is an abductor, is innervated by the first branch of the recurrent laryngeal nerve emerging from within the intrinsic laryngeal muscles [20]. This motor unit is composed of more nerve fibers than the unit that supplies other laryngeal muscles [21]. In a study wherein transected recurrent laryngeal nerves in rats were anastomosed end-to-end, Pitman et al [22] reported that in the early phase of re-innervation, abduction was possible due to re-establishment of contact of several axons with the posterior cricoarytenoid muscle, which is an abductor; but subsequent enhanced innervation of the adductors, which are antagonists, may lead to the disappearance of vocal fold movements. This phenomenon is called "misdirection". This mechanism may have occurred in this study, based on the histological regeneration of nerves and the occurrence of evoked potentials. In future, studies in which electromyograms that analyze the abductors and adductors separately must be performed.

Another conceivable explanation for the temporary restoration of vocal fold movements in this study could be the progressive atrophy of intrinsic laryngeal muscles. As stated earlier, all the examined rats in the 1-mm, 3-mm, and 5-mm groups demonstrated atrophy of the thyroarytenoid muscles at 15 w. If adequate nerve fibers have not been regenerated, intrinsic muscle atrophy is suspected to progress even in the presence of re-innervation. Even after early restoration of vocal fold movements, subsequent progression of atrophy could have hindered the vocal fold movements. However, in this study, vocal fold movements in rats that were restored by 15 w did not show milder atrophy than that in other rats.

Regeneration of a transected recurrent laryngeal nerve and restoration of vocal fold movements to a near-normal level requires recurrent laryngeal nerve regeneration without misdirection and prevention of atrophy of the intrinsic laryngeal muscles. These objectives could be fulfilled by provision of nutrients and cells that promote early regeneration, a tube material and conduit that does not inhibit nerve regeneration, and some means of preventing misdirection.

## Conclusion

In this study, we successfully regenerated the recurrent laryngeal nerves and produced a rat model of recurrent laryngeal nerve regeneration with nerve gaps of 1, 3, and 5 mm that demonstrated temporary recovery of vocal fold movements by bridging the transected recurrent laryngeal nerves with silicon tubes containing type I collagen gel. The rat model could serve as a new means of assessing different types of nerve guidance conduits, cell therapy, and other novel treatments targeted to regenerate nerves.

## Supporting information

**S1 Table.**
(DOCX)

**S2 Table.**
(DOCX)

## Author Contributions

**Conceptualization:** Ryohei Asai, Hiroumi Matsuzaki, Takeshi Oshima, Taro Matsumoto.

**Data curation:** Ryohei Asai.

**Formal analysis:** Ryohei Asai.

**Funding acquisition:** Sohei Ishii, Taro Matsumoto.

**Investigation:** Ryohei Asai, Ikuo Mikoshiba, Tomohiko Kazama.

**Methodology:** Hiroumi Matsuzaki.

**Project administration:** Sohei Ishii, Taro Matsumoto.

**Resources:** Ryohei Asai, Sohei Ishii, Tomohiko Kazama, Taro Matsumoto.

**Supervision:** Hiroumi Matsuzaki, Takeshi Oshima, Taro Matsumoto.

**Validation:** Ikuo Mikoshiba.

**Visualization:** Ryohei Asai, Taro Matsumoto.

**Writing – original draft:** Ryohei Asai.

**Writing – review & editing:** Taro Matsumoto.

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
