## [Decision Letter · Decision Letter 0]

23 Jul 2020

Functional recurrent laryngeal nerve regeneration using a silicon tube containing a collagen gel in a rat model

PONE-D-20-04831

Dear Dr. Asai,

First, I hope that this email finds you well and that you are able to manage your work and home life during the pandemic.  Second, I apologize for the delay in processing your work; finding suitable reviewers with time to commit to the review process during the pandemic proved to be challenging.  Despite these challenges, I am please to inform you that a reviewer and myself have judged this work to be scientifically suitable for publication and will be formally accepted for publication once it meets all outstanding technical requirements.

Kind regards,

Brenton G. Cooper, Ph.D.

Academic Editor

PLOS ONE

Additional Editor Comments (optional):

Reviewers' comments:

Reviewer's Responses to Questions

**Comments to the Author**

1. Is the manuscript technically sound, and do the data support the conclusions?

Reviewer #1: Yes

2. Has the statistical analysis been performed appropriately and rigorously? 

Reviewer #1: Yes

3. Have the authors made all data underlying the findings in their manuscript fully available?

Reviewer #1: Yes

4. Is the manuscript presented in an intelligible fashion and written in standard English?

Reviewer #1: Yes

5. Review Comments to the Author

Reviewer #1: There is a need for a more translational research and models on recurrent laryngeal nerve injury. This novel research helps fill that need. Much work has previously done in cats or larger animal models due to limitations inherent in the rat model, but the authors demonstrate good results using histological and functional outcomes measures. The findings do meaningfully differ between the different gap lengths. The finding of temporary recovery of function followed by subsequent loss of function likely represents onset of laryngeal synkinesis. The quality of figures and histological images is good. Use of blinding and external scorers lends credibility.

The main limitation is that of a short nerve gap that can be overcome by spontaneous regeneration at later time points. Particularly attention to timing with respect to assessments will be important for future such investigations. The role of synkinesis is important to address and limitations of model relating to length. The authors deserve praise for important and instructive work in developing this experimental model.

6. PLOS authors have the option to publish the peer review history of their article (what does this mean?). If published, this will include your full peer review and any attached files.

Reviewer #1: Yes: Michael J. Brenner M.D.

---

## [Editor Report · Acceptance letter]

19 Aug 2020

PONE-D-20-04831 

Functional recurrent laryngeal nerve regeneration using a silicon tube containing a collagen gel in a rat model 

Dear Dr. Asai:

I'm pleased to inform you that your manuscript has been deemed suitable for publication in PLOS ONE. Congratulations! Your manuscript is now with our production department. 

Kind regards, 

on behalf of

Dr. Brenton G. Cooper 

Academic Editor

PLOS ONE